# Three-Dimensional (3D) Stereolithographic Tooth Replicas Accuracy Evaluation: In Vitro Pilot Study for Dental Auto-Transplant Surgical Procedures

**DOI:** 10.3390/ma15072378

**Published:** 2022-03-23

**Authors:** Filiberto Mastrangelo, Rossella Battaglia, Dario Natale, Raimondo Quaresima

**Affiliations:** 1Department of Clinical and Experimental Medicine, University of Foggia, 71100 Foggia, Italy; 2Independent Researcher, 71100 Foggia, Italy; rossellabattaglia1994@gmail.com; 3Independent Researcher, 76125 Trani, Italy; darionatale87@gmail.com; 4Department of Civil Engineering, Architecture and Environment, University of L’Aquila, 67100 L’Aquila, Italy

**Keywords:** 3D-printed replica, cone beam computer tomography, tooth transplantation, DICOM-STL, accuracy

## Abstract

After immediate tooth extraction or after alveolar socket healing, tooth transplants are increasingly used for functional restoration of edentulous maxillary areas. Recent studies have shown the periodontal ligament (PDL) viability and the tooth housing time in the adapted neo-alveolus as key factors for transplantation success. During surgical time, 3D stereolithographic replicas are used for fitting test procedures. In this paper, the accuracy of 3D dental replicas, compared with the corresponding natural teeth, is assessed in surgical transplantation. Lamb skulls were selected and submitted to Cone Beam Computer Tomography (CBCT). Scanning information, converted into Standard Digital Imaging and Communications in Medicine (DICOM) and Standard Triangulation Language (STL), was sent to the Volux X-ray Centre for 3D replica printing. After the tooth extractions, all lambs’ incisors were measured with a digital caliber and compared with the 3D replicas. Volume and dimensional error values were evaluated. All replicas showed macroscopically smaller volume (45.54%). Root replicas showed higher variations compared with the crown areas, with several unreplicated apical root areas. The cement–enamel junction tooth area was replicated quite faithfully, and the base area relative error showed 9.8% mean value. Even further studies with a larger number of replicas are needed. Data obtained confirmed high volumes of macroscopic discrepancies with several unreproduced apical root sites. The achieved accuracy (90.2%) confirmed that the 3D replicas cannot be used to reduce the surgical time during transplantation predictable procedures.

## 1. Introduction

The full digital revolution is inducing rapid developments throughout the world of modern dentistry. Integration of digital technologies in surgical treatment has allowed a transition from a two-dimensional to a three-dimensional (3D) clinical approach, and recently, dental clinicians have increased the utilization of digital scanners, cone beam computed tomography, and 3D printers. Digital technology promotes several advantages for practitioners and patients in all common treatments. However, the lack of accuracy and strictness in the stereolithographic model or replicas could lead to the failure of the surgical procedures [1,2,3].

Oral surgery is also developing techniques and procedures to reduce surgical times, reduce invasiveness and complications, and increase patient compliance and treatment success rates. Currently, dental auto-transplantation can be considered a predictable procedure to replace a missing tooth, and as a valid alternative to modern osseointegrated dental implantology. During the last 15 years, several studies evaluated the viability of the periodontal ligament (PDL) fibers as a key factor in increasing the dental transplant success rate [4,5]. In 2012 Andersson et al., explained how periodontal ligament vitality can prevent failure or ankyloses in the tooth transplant procedures [6]. Therefore, during the fitting procedures, the elapsed time from the tooth extraction and its re-placement into the neo-alveolus prepared site plays a crucial role in preserving the PDL integrity and viability. Another crucial factor is the tooth’s anatomical root conformation. Obviously, similar anatomical conformation of the donor element with the extracted tooth reduces the surgical procedure time. However, in the oral cavity, there are no teeth completely equal and several surgical procedures are needed to adapt the maxillary neo-alveolus to the donor root morphology [7]. Conventional dental transplant procedures used the extracted donor tooth as a template to prepare the receiving neo-alveolus site and to verify the dental adaptation, increasing both the extra-alveolar time tooth permanence and risk of failure of the dental transplant procedure [8,9,10]. In recent years, several authors have promoted the use of 3D stereolithographic tooth replicas as a “fit test”, in order to drastically reduce the tooth extra-alveolar time permanence and minimize the graft manipulation [11,12]. Several autotransplantation studies have reported a short extraoral time of the donor tooth resulted in high success and survival rates of 80.0–91.1%, compared with control cases with increased extra-oral time and lower percentages [13]. Nevertheless, several cases revealed limitations of the teeth replicas due to their linear and volume dimensions, if compared with the donor tooth [14].

Therefore, the surgical procedures became more complex and ineffective due to the necessary adaptations without obtaining the expected results [15,16,17,18]. A full digital pre-surgical procedure endorses the cone beam computed tomography (CBCT) of the donor tooth to evaluate the dental volumes, the anatomical characteristics and the root configuration. In this way, after data processing, a printed 3D stereolithographic replica can be used to organize the surgical neo-alveolus site preparation and reduce the surgical variables and risks [12,18]. However, several 3D replica cases revealed more linear discrepancies and smaller size dimensions compared with the donor tooth [19,20].

The aim of the present study was to assess, in a lamb animal model, the 3D dental replica’s accuracy and the mean error value compared with the corresponding natural mono-radicular extracted tooth.

## 2. Materials and Methods

In 2019, at the Clinical and Experimental Medicine Department School of Dentistry of Foggia University, five lamb skulls were selected, and cone beam computer tomography (CBCT) was performed. After radiological scanning, eight mono-radicular incisors were used to record linear and volume data. Data scans were then converted in Standard Digital Imaging and Communications in Medicine (DICOM) and Standard Triangulation Language (STL) and sent to the Volux X-ray Centre for 3D replica printing. After radiological procedures, all incisor teeth were gently, surgically extracted. All specimens were analyzed with a precision caliber and compared with the 3D replicas. Volume and dimensional error values were detected.

### 2.1. Radiological Acquisition and 3D Printing

Cone beam computer tomography (CBCT) technology (NewTom GiANO^®^, Cefla, Imola, Italy) with a Field of View (FOV) of 11 × 8 cm was used and the scan was displayed using NNT-Viewer software. The files obtained were converted in Digital Imaging and Communications in Medicine (DICOM) and Standard Triangulation Language (STL) format and sent to the Volux X-ray Centre (Santarcangelo di Romagna, Rimini, Italy) for 3D tooth stereolithographic replica printing. After the DICOM file was recorded, a pool of three engineers extrapolated each dental element using special Amira-Fei^®^ Software (Thermo Fisher Scientific—Waltham, MA, USA) and converted it to an STL file (Figure 1).

In the digital file reconstruction, the overlapping and the dental anomalous inclination were analyzed, and a single block of teeth connected by the crowns was evaluated. Single crowns and roots were then separated and the portion of the cement–enamel junction was detected. All crowns were digitally cut in a sagittal way at the interproximal level preserving the teeth landmarks. The digital files obtained for all incisors were sent to a stereolithographic printing machine, DWS DIGITALWAX 020D^®^ (professional resin 3D printer from DWS Additive Manufacturing—DWS-020D Aniwaa Pte Ltd., Singapore) and 3D tooth replica models were produced using polymerized fluid resin (Polyamide 6).

Mesio-distal measurements on the midpoint of the mesial face, and buccal-lingual measurements on the midpoint of the distal face, of each tooth replica were carried out with a digital caliber (TACKLIFE DC02— Plainview, NY, USA) and all diameters were evaluated. All distances between the midpoints of the mesial, distal, buccal, and lingual faces and the apex were also measured in triplicate and the arithmetic means was obtained for each replica.

### 2.2. Tooth Extraction and Samples Measurement

Eight dental elements were selected for each lamb skull. After the radiological procedures were carried out, the lamb mono-radicular incisors were gently extracted (Figure 1b) and immersed in 0.5% sodium hypochlorite solution for 1 h. All samples were then gently cleaned and the fibrous periodontal ligament and bone residues were removed. All incisors were stored in a 0.9% weight/volume NaCl physiological solution. Mesio-distal measurements on the midpoint of the mesial face, and buccal-lingual measurements on the midpoint of the distal face, of each tooth were carried out with a digital caliber (TACKLIFE DC02, Plainview, NY, USA) and all diameters were evaluated (Figure 2).

For each tooth all distances between the midpoints of the mesial, distal, buccal and lingual faces and the apex were measured in triplicate and the arithmetic means were calculated. In order to obtain an assimilated geometric model of the natural tooth and replicas, a rhombic-based pyramid model for the root was adopted (Figure 3).

The root volume was calculated as a product of the mesio-distal (MD) and buccal-lingual (BL) diagonals (base area of the rhombus) by the height of the pyramid (*h* coincides with the longitudinal axis of the tooth/replica) as:(1)$$V=\frac{1}{3}\;[\left(MD\right)\left(BL\right)h]$$

Distances between four points to the cement–enamel junction (CEJ) and the apex (A), AM (mesial point apex), AD (distal point apex), BA (buccal point apex) and LA (lingual point apex) were evaluated with a digital caliber.

To calculate the height of the pyramid on the buccal-lingual plane, the triangular sections (ACM and ACD triangles) were considered. The height was calculated, according to Pythagoras’ theorem, as segment AC, where C is the intersection point of MD and BL diagonals (Figure 3b), as:(2)$$AC=\sqrt{AM^{2}-\frac{1}{2}MD^{2}}$$

In the same way on the mesio-distal plane the height of the pyramid, taking into account the triangles ACV and ACL (Figure 3b), was calculated as:(3)$$AC=\sqrt{AV^{2}-\frac{1}{2}VL^{2}}$$

The two heights obtained were mathematically averaged and the value replaced in the formula Equation (1) to calculate the volume.

Linear and volume measurement percentage errors were analyzed between the natural tooth and the replicas using the equation:(4)$$\mathrm{Linear}\;\mathrm{Er}\%=\frac{Measurements\;on\;natural\;tooth-Measurements\;on\;3D\;model}{Measurements\;on\;natural\;tooth}\;100$$
(5)$$\mathrm{Volume}\;\mathrm{Er}\%=\frac{\mathrm{Natural}\;\mathrm{tooth}\;\mathrm{volume}-3D\;\mathrm{model}\;\mathrm{volume}}{\mathrm{Natural}\;\mathrm{tooth}\;\mathrm{volume}\;}\;100$$

## 3. Results

All 3D replicas appeared smaller than the natural teeth (Figure 4) and showed linear and volume error percentages (Table 1).

The CEJ Mesio/distal and buccal/lingual mean value error were 0.04 and 0.06 with a 4.17 (SD ± 0.0025) and 6.19 (SD ± 0.09) percentage, respectively. The root mean value linear error was higher in all specimens with 0.39 in M/A and D/A, 0.43 in B/A, and 0.40 in L/A measurements. The relative percentage root error was 39.42 (SD ± 0.11) in M/A, 39.22 (SD ± 0.099) in D/A, 43.11 (SD ± 0.09) in B/A, and 39.92 (SD ± 0.087) in L/A. (Table 1) In the CEJ portion, all replicas showed macroscopically good features and a low discrepancy value compared with the natural tooth with 9.8% (SD ± 0.083) error mean value. (Table 2) Relatively shorter root volume was macroscopically evaluated in all replicas compared with the natural tooth, and several apex zones showed high discrepancies and low accuracy grades, with the lack of several features. The replica’s volume showed 0.46 mean value error with a 45.54% (SD ± 0.082) (Table 3).

## 4. Discussion

In 1956, Hale et al., described autogenous molar transplantation clinical technique [21]. Several manuscripts with descriptions of surgical techniques and clinical case reports of mono-radicular vs. pluri-radicular transplanted teeth have been reported [22]. Compared with dental implants, the tooth transplantations showed important advantages to periodontal ligament and proprioception preservation cheapness, and finally it could be performed in young patients in which an implant could not be performed [23]. The risks described were ankylosis, root resorption, and transplantation failure [22,23,24,25]. In literature, two different tooth transplantation surgical techniques were described: In the first, the tooth atraumatic placement was immediately realized into a fresh alveolar socket after the tooth extraction. In the second technique, the surgeon used a surgical burs action to create a new alveolar socket site into the maxillary bone before tooth atraumatical placement [22,23]. In literature, biomaterials and membrane were described, to close the 2/3 mm gap between the transplanted tooth and the receiving bone site [26]. In 2001, Czochrowska et al., after 35 years of surgical procedures, showed a 90–92% success rate in transplanted procedures [27], whereas in 2008, Reich et al., and Jonsson described heterogeneous results during follow-up time observation [22,28]. In 2020, Boschini, L et al., observed an 89.69% success rate in open apex tooth transplants compared with complete formed apex transplants (80%) [29]. The authors hypothesized these differences since, in open apex transplants, pulp stem cells may maintain a higher differentiation rate, whereas after complete apex formation, the transplants could have an endodontic treatment two weeks later in order to reduce pulp and periodontal complications [23]. In 2018, Jakobsen et al., confirmed root formation and the reduction in traumaticity were the crucial factors to obtaining a high tooth transplanted success rate [25], whereas Kafourou et al., in 2017 described the higher diameter of the apex (1–2 mm) with the potential of revascularization and the rate of the tooth eruption in the maxilla as important key points for the success of the procedure [30]. In 2017, Tang et al., in 23 Chinese patients, showed a high success rate related to the shape of the root, which appeared to be preferable when conical and smooth [31]. In 2017, Strabac et al., in a five-year human cohort study, showed better results in the upper arch compared with the lower arch, perhaps due to the reduced thickness of bone [32]. In 2013, Denys et al., and in 2015, Ronchetti et al., showed a higher negative effect rate in multiple-root teeth (81.6%) compared with mono-radicular teeth (33.8%) [33]—perhaps related to the root anatomy—with more trauma induced by extraction procedures [34]. In order to reduce the intra-and extra-oral surgical time of tooth transplantation, maintain the periodontal tissue vitality, reduce the complications, and increase the procedure success rate, several authors have proposed transplanted tooth 3D replicas from CBCT using a surgical template [23,35,36]. In 2016, Verweij et al., and in 2019, Ezeldeen et al., established that this technique could improve the survival rate up to 92% [1,35]. In 1990, Andreasen et al., recommended keeping tooth extra-oral time to less than 18 min during dental surgical transplantation procedures to achieve periodontal healing [24]. However, in 2019, Wu Y affirmed that 3D replica errors may be related to surgical procedures having a longer duration of up to 3.5–4 min [26]. In 2020, Vinci et al., found similar kinds of inaccuracies between DICOM data and stereolithographic models in an implant guided surgery study [37]. Several factors may be responsible for the lack of the replica’s accuracy, such as areas of shrinkage, thinness of layers, or even angulation of 3D printing [38,39], as well as the voxel CBCT data conversion in STL and DICOM format [40]. Concerning shrinkage, it must be considered that every crystalline and/or amorphous, fluid or melting material could be used for 3D printing. The results of the present pilot study have shown a significant linear and volume lack of accuracy in all 3D replicas compared with the natural tooth; all 3D replicas have reported high discrepancy and low accuracy grade of the apex zone with lack of details and features. The root mean value of the linear errors was 0.39 in M/A and D/A, 0.43 in B/A, and 0.40 in L/A measurements, whereas the relative percentage root error was 39.42 (SD ± 0.11) in M/A, 39.22 (SD ± 0.099) in D/A, 43.11 (SD ± 0.09) in B/A, and 39.92 (SD ± 0.087) in L/A. The replica’s volume showed 0.46 mean value error with a 45.54% (SD ± 0.082) overall error. This overall linear and volumetric error, in addition to the complete lack of multiple details and characteristics of the tooth apex area, plays a crucial role during the autotransplant surgical procedures. It is not possible to prepare an adequate area of bone in which to place the donor tooth and maintain the vitality of the cells of the periodontal ligament, which for the clinician is the crucial objective of the surgical procedure.

The factors that contribute to the low accuracy (distortion and shrinkage) of the 3D replica include the resin used [41], the processing parameters [41,42,43], and object shape [44,45]. The high critical defects macroscopically observed on all the replicas in the root apex zone, as well as the low accuracy grade, seem to be correlated to the resin used and to the digital file reconstruction. Considering that the shrinkage rate of the polyamide polymer used (PA6-Poly(azepan-2-one); poly(hexano-6-lactam)) ranges within 0.5–1.6% [46,47] the main remaining error percentage seem to be ascribed to the accuracy of the complex radiological acquisition (CBCT combined with NNT-Viewer software) and STL conversion procedures.

## 5. Conclusions

Further studies with a greater number of specimens and a comparison of different software and polymeric materials will be needed to confirm the present pilot study. However, the results confirmed that the 3D replicas from CBCT scans of the jaws were not accurate in linear and volume measurements. Currently, the 3D replicas’ high lack of volume accuracy (around 45%) does not indicate their use would contribute to the reduction in surgical times needed to improve the success rate during dental transplant procedures. Indeed, the 3D tooth replicas’ high error rates regarding the apical areas are crucial during the tooth placement procedures into the neo-alveolus site. Further information will have to be evaluated for multi-rooted elements in order to verify whether the margin of error of the 3D replicas recorded in this study will remain constant or will be different in the presence of a further variable given by the complex anatomy of the transplanted multi-rooted tooth.

## Figures and Tables

**Figure 1 materials-15-02378-f001:**
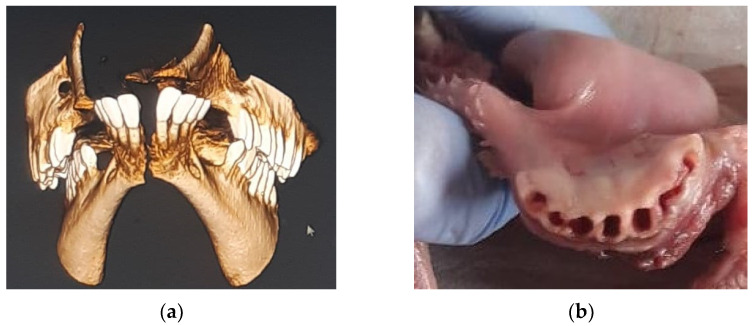
The tridimensional reconstruction of lamb before: (**a**) teeth extraction in DICOM format and (**b**) clinical view of lamb skull after teeth extraction.

**Figure 2 materials-15-02378-f002:**
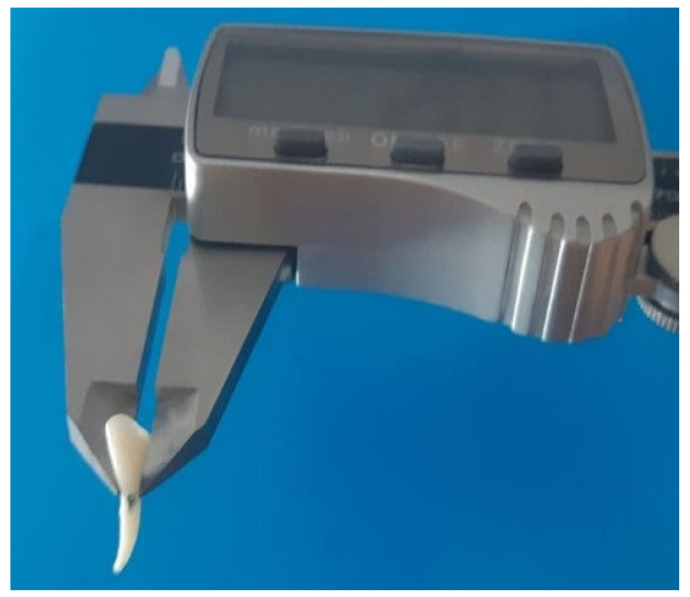
Digital caliber measurements of the mesio-distal and buccal-lingual diameters (hundredths millimeter precision). In the first case the locating branches of the caliber were placed on the midpoint of the mesial face and midpoint of the distal face of the tooth, likewise it was carried out for the measurement of the buccal-lingual diameter.

**Figure 3 materials-15-02378-f003:**
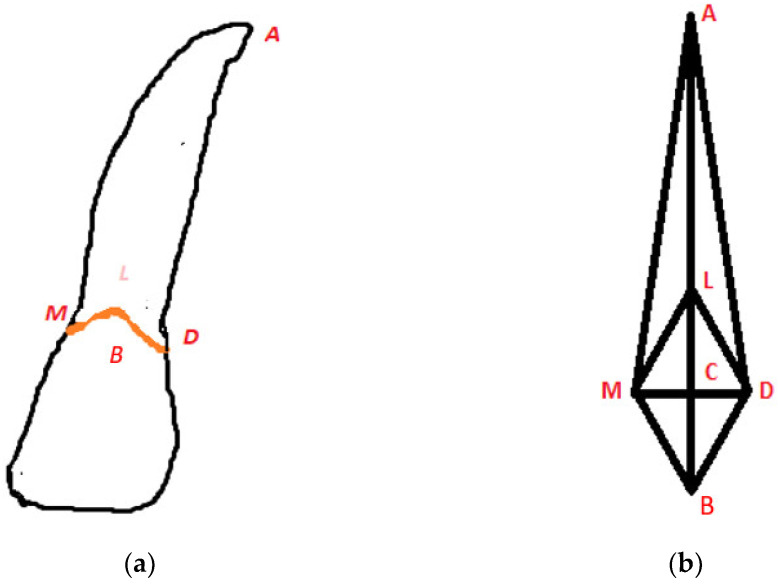
Dental root representation and geometric measurements: (**a**) assimilation to a polyhedric figure with a pyramidal shape as a base (i.e., the crown, delimited by the cement–enamel junction) and (**b**) a rhombus and a vertex that stands for dental apex. Legend: Natural tooth root midpoints: (A) Apex; (M) Mesial; (D) Distal; (B) Buccal. Tooth root replica geometric model: (A) Apex: (MD) Mesial-distal diagonal; (BL) Buccal-lingual diagonal; (C) Intersection point between mesial-distal and buccal-lingual diagonals.

**Figure 4 materials-15-02378-f004:**
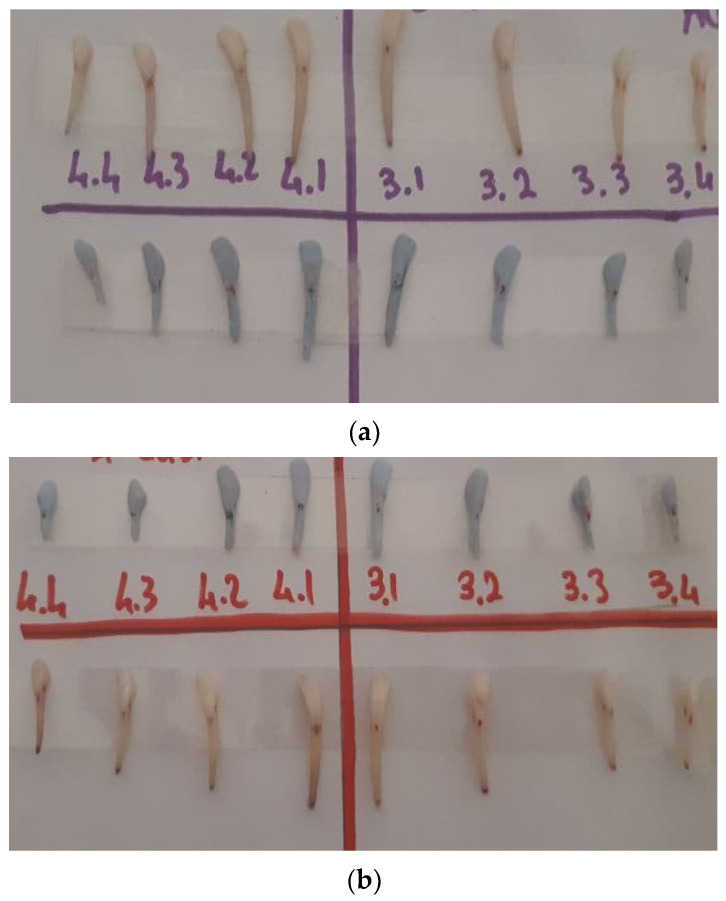
Comparison of natural teeth once extracted and their 3D replicas: (**a**) first lamb skull; (**b**) second lamb skull.

**Table 1 materials-15-02378-t001:** Cement–Enamel Junction (CEJ) and Root percentage error evaluation at mesial, distal, buccal, and lingual points in all natural tooth and replica specimens. All measurements were calculated using the aforementioned formula. Legend: a1 = lamb skull 1; a2 = lamb skull 2; M/A = mesial point and apex; D/A = distal point and apex; B/A = buccal point and apex; L/A = lingual point and apex.

Error Percentage (%)						
	CEJ		Root			
Tooth	Mesio/Distal	Vestib/Lingual	M/A	D/A	B/A	L/A
3.1 a1	0.01	0.04	0.29	0.33	0.37	0.35
3.2 a1	0.04	0.02	0.33	0.37	0.53	0.54
3.3 a1	0.04	0.02	0.39	0.39	0.42	0.40
3.3 a1	0.04	0.07	0.42	0.43	0.53	0.42
4.1 a1	0.05	0.09	0.26	0.27	0.23	0.24
4.2 a1	0.03	0.00	0.39	0.35	0.41	0.33
4.3 a1	0.07	0.01	0.59	0.59	0.41	0.39
4.4 a1	0.01	0.02	0.51	0.53	0.53	0.45
3.1 a2	0.06	0.03	0.26	0.26	0.35	0.30
3.2 a2	0.02	0.36	0.29	0.31	0.40	0.37
3.3 a2	0.05	0.07	0.32	0.39	0.39	0.41
3.3 a2	0.06	0.03	0.32	0.30	0.36	0.35
4.1 a2	0.07	0.03	0.29	0.30	0.34	0.31
4.2 a2	0.02	0.00	0.45	0.37	0.50	0.42
4.3 a2	0.01	0.15	0.56	0.51	0.55	0.55
4.4 a2	0.07	0.00	0.53	0.51	0.52	0.52
Average	0.04	0.06	0.39	0.39	0.43	0.40
Error (%)	4.17	6.13	39.42	39.22	43.11	39.92
Stand Dev	0.02	0.09	0.11	0.09	0.09	0.08

**Table 2 materials-15-02378-t002:** Percentage error calculated through the ratio between the base area of the natural tooth and the prototype.

Error Percentage (%)	
Tooth Number	Basic Area
3.1 a1	0.05
3.2 a1	0.06
3.3 a1	0.05
3.3 a1	0.11
4.1 a1	0.14
4.2 a1	0.04
4.3 a1	0.08
4.4 a1	0.03
3.1 a2	0.09
3.2 a2	0.38
3.3 a2	0.12
3.3 a2	0.09
4.1 a2	0.10
4.2 a2	0.02
4.3 a2	0.14
4.4 a2	0.07
Average	0.098
Error (%)	9.80
Stand Dev	0.083

**Table 3 materials-15-02378-t003:** Assessment of the percentage error calculated by evaluating the difference between the volume of the natural tooth and the prototyped tooth. For problems of a technical nature, the top third of the models was reduced and this justified its high error percentage. (see Figure 4).

Error Percentage (%)	
Tooth Number	Volume
3.1 a1	0.37
3.2 a1	0.48
3.3 a1	0.44
3.3 a1	0.52
4.1 a1	0.38
4.2 a1	0.40
4.3 a1	0.55
4.4 a1	0.53
3.1 a2	0.36
3.2 a2	0.36
3.3 a2	0.46
3.3 a2	0.39
4.1 a2	0.38
4.2 a2	0.42
4.3 a2	0.62
4.4 a2	0.57
Average	0.46
Error (%)	45.54
Stand Dev	0.082

## Data Availability

Data presented in this study are available on request from the corresponding author.

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
