# Peer review of "Three-Dimensional (3D) Stereolithographic Tooth Replicas Accuracy Evaluation: In Vitro Pilot Study for Dental Auto-Transplant Surgical Procedures"

_materials, 2022, doi:10.3390/ma15072378_

Round 1
Reviewer 1 Report
This manuscript discusses the assessment of a 3D dental replica in a lamb animal model. Interesting article that should be publishable with a few minor comments.
1. Please define the letters in the caption for Figure 3.
2. A minor comment, but please do not use the letter x for multiplication symbols in Equation (1).
3. Please use an equation editor for the equations (2)-(4).
Reviewer 2 Report
The authors present a study were they assess, in a lamb animal model, the 3D dental replicas 65
accuracy and the mean error value compared to the corresponding natural mono-radicular 66
extracted tooth.
Better model for the volume calculation needs to be included.
Eq. 1: replace x by dot or star
Eqs. In 2.2. needs to be written in appropriate way and also well explained.
My opinion: more material characteristic needs to be in included. Otherwise the paper will not fit into journal Materials.
Reviewer 3 Report
The paper is of interest to those in the field of dentistry.
It is an interesting study carried out by highlighting some particularly important aspects in the case of this dental technique.
Reviewer 4 Report
This paper is very interesting in terms of indicating the limit of reproductivity of the combined techniques of 3D tomography and 3D printing to the animal body parts.
However, research itself is just a careful quantification of the obvious results. Since resin’s shrinkage (and also measurement error of scanners) is well-known cause of size error, so it would be more meaningful if this manuscript could present a concrete solution. I recommend more detail discussion of the results by the authors.
The discussion is limited to quotations from previous studies, and there is almost no discussion obtained from this study. There is no verification of the materials themselves, even though this is a submission to a materials journal. I think that a discussion from the viewpoint of materials science should be included.  I recommended a minor revision, but to be honest, I would recommend submitting it to a journal specializing in dental materials for clinicians.  I have reviewed other papers in this SI in the past, and when I recommended submission to such journals, since it was only about materials and there was no discussion from the view of material science. However, the paper was accepted at the editor's discretion.  Therefore, I decided that a minor revision since the editor judged that it was OK as long as the subject was dental material. Since the discussion is not really a discussion, I recommend that the discussion be revised anyway.
Round 2
Reviewer 2 Report
The paper can be accepted